# Relationship of spirituality, health engagement, health belief and attitudes toward acceptance and willingness to pay for a COVID-19 vaccine

Sri Handayani[1,2☉], Yohanes Andy Rias[2,3,4☉], Maria Dyah Kurniasari[4¤], Ratna Agustin[5], Yafi Sabila Rosyad[1], Ya Wen Shih[6], Ching Wen Chang[7], Hsiu Ting Tsai[2,4]*

**1** Faculty of Health and Medicine, College of Nursing, Sekolah Tinggi Ilmu Kesehatan Yogyakarta, Yogyakarta, Indonesia, **2** Post-Baccalaureate Program in Nursing, College of Nursing, Taipei Medical University, Taipei, Taiwan, R.O.C, **3** Faculty of Health and Medicine, College of Nursing, Institut Ilmu Kesehatan Bhakti Wiyata Kediri, Kediri, Indonesia, **4** School of Nursing, College of Nursing, Taipei Medical University, Taipei, Taiwan, R.O.C, **5** Faculty of Health and Medicine, College of Nursing, Universitas Muhammadiyah Surabaya, Surabaya, Indonesia, **6** School of Nursing, National Taipei University of Nursing and Health Science, Taipei, Taiwan, R.O.C, **7** Department of Obstetrics and Gynecology, Taipei Medical University Hospital, Taipei, Taiwan, R.O.C

☉ These authors contributed equally to this work.
¤ Current address: Faculty of Medicine and Health Science, Nursing Department, Universitas Kristen Satya Wacana, Salatiga, Indonesia
* tsaihsiuting@yahoo.com.tw, hsiuting@tmu.edu.tw

**Data Availability Statement:** Data are available without restriction: S1 Data Set. doi//10.6084/m9.figshare.20400603 S1 File. https://doi.org/10.6084/

## Abstract

### Purpose

To explore the wider determinant factor of citizens' spirituality, health engagement, health belief model, and attitudes towards vaccines toward acceptance and willingness to pay for a Coronavirus disease 2019 (COVID-19) vaccination.

### Methods

A community-based cross-sectional online investigation with convenience sampling was utilized to recruit 1423 citizens from 18 districts across Indonesia between December 14, 2020 and January 17, 2021. Descriptive statistics, One-way analysis of variance, Pearson correlation, Independent t-tests, and multiple linear regression were examined.

### Results

Spirituality, health engagement and attitude toward vaccines, as well as health beliefs constructs (all scores of perceived benefits and barriers) were significant key factors of acceptance of vaccines. Interestingly, the spirituality, attitude toward vaccine, and health beliefs constructs including perceived susceptibility, and benefits indicated a significantly higher willingness.

### Conclusions

Results demonstrated the utility of spirituality, health engagement, health belief model, and attitudes towards vaccines in understanding acceptance and willingness to pay for a

m9.figshare.20424897.v2 S1 Table. https://doi.org/
10.6084/m9.figshare.20424909.v1 S2 Table.
https://doi.org/10.6084/m9.figshare.20425041.v1
S3 Table. https://doi.org/10.6084/m9.figshare.
20425065.v1 S4 Table.https://doi.org/10.6084/m9.
figshare.20425020.v1.

**Funding:** This research was funded by the Ministry
of Science and Technology (MOST), Taiwan,
through grant nos. 106-2314-B-038-013-MY3 and
109-2314-B-038-110-MY3 by H.T.T. The funders
had no role in study design, data collection and
analysis, decision to publish, or preparation of the
manuscript.

**Competing interests:** The authors have declared
that no competing interests exist.

vaccine. Specifically, a key obstacle to the acceptance of and willingness to pay COVID-19 vaccination included a high score of the perceived barrier construct. Moreover, the acceptance of and willingness to pay could be impaired by worries about the side-effects of a COVID-19 vaccination.

## Introduction

Coronavirus disease 2019 (COVID-19) caused clusters of a complex respiratory syndrome characterized with a novel beta-coronaviruses infection [1]. As of May 31, 2021, the world health organization confirmed that 170,051,718 individuals had been infected with COVID-19 worldwide [2]. From 3 January 2020 to 13 June 2021, this disease has spread to Indonesia, where approximately 1,816,041 people are reported to be infected, with 50,404 deaths [3]. After the viral sequence (severe acute respiratory syndrome coronavirus 2) was published, vaccines for COVID-19 were rapidly developed to be distributed globally [4, 5]. While vaccine programs could substantially alleviate the spread of the virus, one of the problems for policy-makers is determining how to motivate their citizens to get vaccinated. Most vaccine skeptics refuse to be vaccinated [6]. Interestingly, Indonesia is unique because its citizens have higher positive spiritual beliefs related to health attitudes [7] and differences in health perspectives [8], which may influence acceptance and willingness to pay for the COVID-19 vaccine.

Acceptance and willingness to pay for a COVID-19 vaccine are critical to the success of a high-coverage vaccination program [9, 10]. Recent studies showed that the acceptance of vaccination COVID-19 in the United States [11], Russia [12], Malaysia [9], and Jordan [13] were 67%, 55%, 48.2%, and 37.4%, respectively. Moreover, an epidemiological study in low- or middle-income countries such as Bangladesh, India, Iran, Pakistan, Egypt, Nigeria, Sudan, Tunisia, Brazil, and Chile presented that the acceptance of vaccination was approximately 58.3% to 80.1% [14]. A global survey showed that differences in acceptance of a vaccination ranged more than 70% among citizens in 19 countries [12]. The majority of the population in Malaysia [9], Chile [15], Ecuador [16], and the US [17] indicated willingness to pay for a vaccination. Previous studies among Indonesian citizens reported that the vaccine acceptance was 93.3% [18] and 78.3% willingness to pay a vaccination COVID-19 [19]. Nevertheless, these studies only concerned socioeconomics, pre-existing susceptibility to COVID-19-related facts, and risk perception variables [18, 19].

Spirituality as a therapeutic approach for healthcare systems plays a critical role in encouraging healthy behaviors using the power of faith and beliefs [20]. Also, spirituality is an adaptation in response to the transmission dynamics of COVID-19 [7, 21]. Ancient wisdom from spiritual fields can be very useful in encouraging citizens to survive the threat of the COVID-19 pandemic [22]. Indonesia is unique because citizens typically have strong positive spirituality linked to health behaviors and health beliefs [7, 23]. Whereas studies of spirituality have seldom been linked to adaptive response, research on spirituality and medical status has been largely [7, 21, 22], but not exclusively focused on willingness to pay and acceptance a COVID-19 vaccine. Thus, investigating spirituality might be a beneficial valuable approach to promote new insights into implementing vaccination programs.

Remarkably, health engagement (HE) is characterized as private proactivity in the administration of wellbeing related concerns [24] and can improve health behavior [25–27]. However, lack of study to investigate the association between HE and vaccine acceptance [24]. Previous study in Italy presented that vaccine attitudes (AVs) strongly correlated with acceptance a COVID-19 vaccine [24]. Additionally, high scores of HE has been significantly associated with

attitude toward against COVID-19 [28]. Consequently, a high level of citizens' health engagement with high vaccine acceptance seems crucial in the case of an COVID-19, as it is a beneficial premise to guarantee the effectiveness of immunization and spread prevention measures of COVID-19 [24].

Health belief model (HBM) might predict health promoting behaviors in terms of belief patterns by understanding the interaction between health behaviors and health services utilization [29, 30]. Previous studies revealed that HBM was significantly associated with acceptance and willingness a COVID-19 vaccine [31, 32]. Research from Malaysia proved that citizens who believed in the perceived benefits of a vaccination had a positively stronger 2.51-fold risk of acceptance, and the higher score of perceived severity was correlated with a higher level of willingness to pay for a vaccine [9]. However, few researches have explored the various constructs of the HBM that could predict the acceptance of the COVID-19 vaccine, although researchers have analyzed the acceptance of and willingness to pay for the COVID-19 vaccine [31–33].

Immunization campaigns are indeed considered effective when vaccination programs have a significantly high rate of vaccine acceptance [9, 11] and a willingness to pay for it [9]. Interestingly, no study has been conducted on COVID-19 vaccine acceptance and willingness to pay with specific determinations factors such as spirituality, HBM, HE and AVs in Indonesia. To fill these gaps, this study explored how Indonesians accepted the COVID-19 vaccine and their willingness to pay for it. This was accomplished by surveying their spirituality, HE, HBM constructs, and AVs.

## Methods

### Design and participants

A cross-sectional online survey-based overview during COVID-19 for 18 provinces out of 34 provinces in Indonesia. All the information was gathered utilizing snowball sampling technique. The eligible target population was Indonesian citizens aged between 17 and 65 years old, who understood Bahasa Indonesia, currently stay in Indonesia, and filled the consent form. Citizens who had previously been confirmed with suspected COVID-19 were excluded. The research was administered and reported based on the Strengthening the Reporting of Observational Studies in Epidemiology (STROBE) protocol (S1 Checklist).

### Data collection procedure

The online survey was distributed using a Google Form link that was shared on social media platforms including WhatsApp, Instagram, Telegram, and Facebook. Furthermore, this relies on researchers' technical and personal networks and engaging with and distributing the survey through social media influencers and community leaders. Participants were selected for the study using a simplified snowball sampling technique, and they were asked to forward the invitation to their contacts; the estimated completion time for the survey was 15 minutes. We conducted different procedures to target as many respondents as possible from across the region during the December 15, 2020 to January 12, 2021 data collection period. Finally, 1,423 people responded to our Google form. We used participants' email to avoid overlapping responses during data collection.

The Google Form link had four sections. (1) Before allowing participants to proceed to the survey questions, the first section informed them of the objective of the study and eligibility requirements. Furthermore, the informed consent was taken by checking the box "Agree," which was required to confirm that they understood the authorization information and met the inclusion and exclusion criteria. Additionally, participants decided to participate

voluntarily and with the freedom to withdraw at any time; (2) Second section comprised questions correlated to sociodemographic; (3) Third section comprised questions that assessed the intention to accept being vaccinated and willingness to pay for vaccinated; (4) Fourth section contained 35 questions including HE, AVs, HBM, and spirituality questionnaire. Finally, a page at the end expressed our gratitude, and all individuals who completed the survey were encouraged to persuade new respondents from their contact lists to participate by forwarding the link to the online survey.

## Measurements

Participants were instructed to fulfill the online sociodemographic questionnaire consisting of information on age, income, gender, educational levels, geographical region, marital status, urbanicity, and the pandemic's impact on their income.

Intention to accept the vaccine was assessed using a one-item question: "Do you intend to accept vaccination for COVID-19?" with response as continuous data of vaccine acceptance on five-point Likert scale; 1 (not likely at all) to 5 (absolutely). This instrument was adapted from previous studies [24]. The total score ranges from 1 to 5, a higher score indicates a more-favorable attitude to acceptance a COVID-19 vaccine.

Willingness to pay for a vaccination was evaluated using a one-item question: "Are you willing to pay US$17.70~35.40 for a vaccination COVID-19?". Response this statement was ranked on a five-point Likert scale; 1 (not likely at all) to 5 (absolutely). This instrument was adapted from previous studies [19, 24]. The total score ranges from 1 to 5, the higher the score an individual has, the greater their willingness to pay for a vaccine.

The Daily Spiritual Experiences Scale (DSES) contains sixteen questions regarding their spirituality on a 6-point Likert scale; 1 = at no time to 6 = many times a day [34]. Previous study revealed that the Indonesian translation version of the DSES questionnaire-spirituality had Cronbach's alpha of 0.88 [7]. The internal consistency reliability was calculated by Cronbach's alpha test; a value $\geq 0.70$ indicates acceptable reliability [35]. In our study, the Cronbach's alpha value for spirituality was 0.70. The total score ranges from 16 to 96, the greater the number of experiences points a person has, the greater their spirituality. Participants' overall spirituality was categorized, as high if the score was $\geq 72$, and low if the score was $< 72$ [7].

In the present study, the questionnaires including HE, AVs, and HBM were assessed for the translation process. After obtaining approval from the original authors, the questionnaires (HE, AVs, and HBM) were independently translated into Indonesian using the forward and back-translation methods. The questionnaires were translated by five translators, a certified translator and four experts in nursing research in Indonesian universities, whose native language was Indonesian and who were bilingual and fluent in English. The translators were assessing the questionnaire items to be relevant to measure the HE, AVs, and HBM toward acceptance and willingness to pay a COVID-19 vaccination precisely for linguistic and conceptual equivalence. In brief, Indonesian-speaking academics were first contacted to review the translated version for grammatical accuracy and clarity. Thus, four independent bilingual translators completed the back translation of the Bahasa edition into English. In addition, the final Bahasa version was obtained by comparing the original questionnaire with its back translation. Translators were instructed to avoid metaphors, colloquial terminology, and hypothetical statements, and to use simple sentences. Initially, prior to completing the formal online survey, we conducted a pilot study with 60 residents in the close surroundings of the researchers to determine the questionnaire's readability and reliability". Further, we reviewed cognitive debriefing results and the finalized version with content validity index (CVI) and kappa ($k^*$). Finally, we conducted an analysis of the reliability and validity with the Kaiser-Meyer-Olkin

(KMO) test, the Bartlett's test of sphericity value, Cronbach's alpha and item-total correlation coefficient.

Health engagement (HE) consists of six questions with a 5-point Likert scale ranging from 1 (strongly disagree) to 5 (strongly agree). The total score ranges from 6 to 30, a greater value indicating greater HE [24]. Interestingly, we defined HE score with response as continuous data on five-point Likert scale; 1 (definitely disagree) to 5 (strongly agree). Also, we defined HE scores as categorical data for disagreement (definitely disagree/disagree/strongly disagree) and agreement (agree/strongly agree) presented in S1 Table. In our study, HE questionnaire English was translated into Indonesian and had a CVI of 0.93, $k^*$ of 0.94 to 1, the value of the KMO test was 0.72 and the Bartlett's test of sphericity value was significant ($p < 0,001$). Furthermore, Cronbach's alpha of 0.91 with item-total correlation coefficient score was 0.68 to 0.88.

Vaccine attitudes (AVs) consist of two questions with a 5-point Likert scale ranging from 1 (strongly disagree) to 5 (strongly agree). The total score ranges from 2 to 10, a greater value indicating greater AVs. For our study analysis, we defined AVs score with response as continuous (total score). Also, we defined VAs scores involving the agreement (strongly agree/agree), and disagreement (neither agree nor disagree, strongly disagree/disagree) presented in S1 Table. The questions are as follows; (1) "COVID-19 vaccination could have serious collateral effects on my own health"; and (2) "I am sure of the vaccine's effectiveness in preventing infectious diseases such as COVID-19" [24]. The Indonesian version of the VAs questionnaire had an acceptable CVI 0.95 with $k^*$ of 0.98 to 1. The value of the KMO test was 0.69 and the Bartlett's test of sphericity value was significant ($p < 0,001$). Furthermore, a total Cronbach's alpha of 0.70 with item-total correlation coefficient score was 0.60 and 0.68 in our study.

HBM constructs a section which included perceived benefits (PBE), susceptibility (PSU), barriers (PBA), and severity (PSE) and consists of 12 items questions. Response this statement was ranked on a 7-point Likert-scale; 1 (strongly disagree) to 7 (strongly agree) [36]. Also, HBM constructs were used in COVID-19 vaccinations previous research [9, 32]. The total score ranges from 12 to 84, a higher score indicates a good health belief, except for the PBA construct. In the present study, we defined HBM score with response as continuous data or total score in each construct. Moreover, the detailed HBM constructs score involve the agreement (somewhat agree/agree/strongly agree), and disagreement (somewhat disagree/disagree/ strongly disagree /neither agree nor disagree) presented in S2 Table. In our study, the questionnaire of the HBM Indonesian version presented that the CVI was 0.95 with $k^*$ of 0.89 to 0.92. The value of the KMO test was 0.61 and the Bartlett's test of sphericity value was significant ($p < 0,001$). Furthermore, the total of Cronbach's alpha of 0.81 with item-total correlation coefficient score was 0.63 to 0.71.

## Sample size and power calculation

Sample size was estimated based on previous study [37] with the formula; $n = u_a p (1 - p)/\delta^2$, where n = minimum desired sample size, $u_a$ = the standard normal deviation, usually set as 1.96 which corresponds to 5% level of significance. $p$ = the average rate of acceptance of vaccine was estimated on the basis of the available literature and its value was set at 85% [38], $\delta$ = of precision set at 0.015. The calculated minimum sample size was 1,111 (n = 1.96 x 0.85 x (1–0.85)/$0.015^2$ = 1,111). We expected a potential missing data of 20% with a large population and thus aimed to recruit at least 1,388 participants. Finally, during one-month data collection, the total sample consisted of 1,423 Indonesian citizens.

The sample size was calculated based on estimates from the distribution of the general population as reported by the Central bureau of statistics, Indonesia. Proportions from eastern,

central and western regions of Indonesia are reported at 2.76%, 16,14% and 81.10% respectively [39]. In our study, we reached participants from all regions of Indonesia and obtained 11.9%, 16.9% and 71.2% from each base, which has a similar pattern to the proportional distribution of these regions in the general population.

### Data analysis

Descriptive analysis statistics was utilized to evaluate demographic, HE, AVs, health beliefs, and spirituality between groups. The findings are reported as percentages (%) and frequencies (n). The mean and standard deviation (SD) of continuous variables were calculated using an independent t-test, or one-way analysis of variance (ANOVA) and Pearson correlation. The variance inflation factor (VIF) of 10 was used to calculate multicollinearity [40]. This study had a maximum VIF of 4.946 for vaccination acceptance and of 4.996 for willingness to pay for a vaccine, s demonstrates that the results had a low level of multicollinearity. Absolute values for skewness and kurtosis were used to assess normality of the data; skewness value of -0.264 and kurtosis value of 1.677 indicated a normal distribution [41]. Multiple linear regression was used to obtain adjusted beta coefficients (β) with 95% confidence intervals (CIs) for willingness to pay and acceptance a vaccine, and were correlated to exposures of interest (HE, AVs, HBM, and spirituality) after adjusted for potential covariate factors. For all statistical analyses, SPSS Vers. 25 IBM (Armonk, NY, USA) was utilized, and statistical significance was defined as $p < 0.05$.

### Ethical considerations

All study protocols were approved by the Ethics Committee of Institut Ilmu Kesehatan Strada Indonesia (Reference No.: 2228/KEPK/XII/2020). Informed consent was granted from each respondent online who was assured of anonymity and confidentiality, their freedoms to withdraw from the study whenever and that the input information were collected for academic use only.

### Results

Table 1 presented that there were significant differences by geographical region in vaccine acceptance and willingness to pay for the vaccine. However, we found no significant difference in marital status or education level of vaccine acceptance and willingness to pay. Willingness to pay for the vaccine also significantly differed by gender, age, income, urbanicity, and the pandemic's impact on their income, but not in vaccine acceptance among Indonesian citizens.

We presented specific correlation spirituality, HE, AVs and HBM constructs variables for vaccine acceptance and willingness to pay for the vaccine in Table 2. Not only spirituality (r = 0.179, $p < 0.001$) but also, HE (r = 0.718), AVs (r = 0.677), and PSU (r = 0.335), PSE (r = 0.316), PBE (r = 0.392) were positively correlated with acceptance a vaccine. Moreover, HE, AVs, and all HBM constructs were positively correlated with willingness to pay for a vaccine. Contrarily, spirituality (r = −0.057), and PBA (r = −0.086) were negatively correlated with willingness to pay for a vaccine.

Table 3 summarizes the findings of the multiple linear regression performed of the overall score of spirituality, HE, HBM constructs, and AVs for vaccine acceptance and willingness to pay. A higher spirituality, increases the coef. β of having the acceptance to vaccinate against COVID-19 (Adjusted coef. β = 0.01, 95% CI = 0.01~0.02). However, a higher spirituality, declines the coef. β of having the willingness to pay a vaccine (Adjusted coef. β = -0.01, 95% CI = -0.02~-0.01). Moreover, higher HE, AVs, and PBE increases the coef. β of having the

**Table 1. Relationships of distributions of demographic and determinant factors with acceptance of and willingness to pay for COVID-19 vaccine (n = 1423).**

| Variable | All participants (n = 1423) n (%) | Acceptance | | Willingness to pay | |
|---|---|---|---|---|---|
| | | Mean (SD) | p value | Mean (SD) | p value |
| Gender [a] | | | | | |
| Men | 602 (42.3) | 3.85 (1.08) | 0.332 | 2.69 (1.28) | 0.006 |
| Women | 821 (57.7) | 3.90 (0.92) | | 2.51 (1.09) | |
| Age (years) [b] | | | | | |
| 17~24 | 640 (45.0) | 3.84 (1.01) | 0.216 | 2.50 (1.13) | 0.047 |
| 25~39 | 550 (38.7) | 3.94 (0.93) | | 2.66 (1.21) | |
| >40 | 233 (16.4) | 3.88 (1.09) | | 2.64 (1.20) | |
| Marital status [a] | | | | | |
| Not married | 861 (60.5) | 3.90 (0.97) | 0.380 | 2.57 (1.17) | 0.598 |
| Married | 562 (39.5) | 3.85 (1.02) | | 2.60 (1.19) | |
| Education [a] | | | | | |
| ISCED <3 | 51 (3.6) | 3.84 (0.90) | 0.772 | 2.29 (1.10) | 0.072 |
| ISCED ≥3 | 1372 (96.4) | 3.88 (0.99) | | 2.60 (1.18) | |
| Income (IDR) [b] | | | | | |
| <2.5 million | 782 (55.0) | 3.89 (0.98) | 0.585 | 2.46 (1.13) | <0.001 |
| 2.5~5 million | 432 (30.4) | 3.88 (0.95) | | 2.72 (1.15) | |
| 6~10 million | 166 (11.7) | 3.82 (1.09) | | 2.63 (1.21) | |
| >10 million | 43 (3.0) | 4.05 (1.21) | | 3.30 (1.64) | |
| Geographical region [b] | | | | | |
| Western region | 1013 (71.2) | 3.87 (0.97) | 0.013 | 2.48 (1.12) | <0.001 |
| Eastern region | 169 (11.9) | 4.08 (1.01) | | 1.28 (0.10) | |
| Central region | 241 (16.9) | 3.80 (1.04) | | 1.24 (0.08) | |
| Urbanicity [a] | | | | | |
| Rural | 635 (44.6) | 3.91 (0.95) | 0.404 | 2.51 (1.17) | 0.028 |
| Urban | 788 (55.4) | 3.86 (1.02) | | 2.65 (1.18) | |
| Pandemic's impact on income [a] | | | | | |
| No impact | 733 (51.5) | 3.88 (0.99) | 0.831 | 2.74 (1.19) | <0.001 |
| With an impact | 690 (48.5) | 3.89 (0.99) | | 2.42 (1.14) | |

Data are presented as the mean ± standard deviation (SD), or frequency and percentage. COVID-19 = coronavirus disease 2019, ISCED = international standard classification of education; IDR = Indonesian rupiah.

[a] an independent $t$-test or;

[b] a one-way ANOVA.

$p<0.05$ indicates statistical significance

acceptance to vaccinate, but higher PBA score declines the coef. β of having the acceptance to vaccinate.

Additionally, we observed that the means (SD) of vaccine acceptance and willingness to pay were significantly higher for citizens who agreed with all statements of HE, and AVs. Moreover, a higher score of acceptance was identified in citizens with a higher spirituality score of ≥72, but a willingness to pay was not significant as outlined in S1 Table. We also observed that the means (SD) of vaccine acceptance and willingness to pay were significantly higher for citizens who agreed with health beliefs constructs (PSU, PSE, and PBE). In analyzing citizens' PBA, those who reported disagreeing with the statements "the side-effects of vaccination interfere with my usual activities" (PBA1), "I am scared of needles" (PBA2), and "I cannot be bothered to get a vaccination" (PBA3) were positively associated with high vaccine acceptance

**Table 2. Correlation of citizen's spirituality, health engagement, and attitudes with their acceptance and willingness to pay for the COVID-19 vaccine (n = 1423).**

| Variables | All participants | Acceptance | | Willingness to pay | |
|---|---|---|---|---|---|
| | Mean (SD) | r | p value | r | p value |
| Spirituality | 72.32 (8.15) | 0.179 | <0.001 | -0.057 | 0.031 |
| Health engagement (HE) | 23.12 (4.33) | 0.718 | <0.001 | 0.269 | <0.001 |
| Attitudes towards vaccines (AVs) | 6.87 (1.70) | 0.677 | <0.001 | 0.266 | <0.001 |
| Health beliefs–Perceived susceptibility (PSU) | 14.14 (4.51) | 0.335 | <0.001 | 0.276 | <0.001 |
| Health beliefs–Perceived severity (PSE) | 14.04 (4.18) | 0.316 | <0.001 | 0.240 | <0.001 |
| Health beliefs–Perceived benefits (PBE) | 14.27 (3.91) | 0.392 | <0.001 | 0.328 | <0.001 |
| Health beliefs–Perceived barriers (PBA) | 11.26 (3.61) | -0.303 | < .001 | -0.086 | 0.001 |

Data are presented as the mean ± standard deviation (SD), correlation and significant value. COVID-19 = coronavirus disease 2019; AVs = attitude towards vaccines; HE = health engagement, PBA = perceived barriers; PBE = perceived benefits; PSE = perceived severity; PSU = perceived susceptibility. $p$ values were calculated using a Pearson correlation; $p<0.05$ indicates statistical significance.

scores, but were not correlated with a willingness to pay, except for PBA3 are outlined in S2 Table.

S3 Table presented the findings of the multiple linear regression performed of all the indicators of spirituality, HE, health beliefs, and AVs for vaccine acceptance and willingness to pay. Further statistical test showed that getting high spirituality was connected with a higher vaccine acceptance value (β = 0.14, 95% CI = 0.07~0.21), and was significantly associated with a willingness to pay (β = -0.25, 95% CI = -0.37~-0.14) among Indonesian citizens. Both all items of HE and AVs were the strongest determining factors of the vaccine acceptance score. However, HE items 1 to 5 and AVs1 were not correlated with a willingness to pay after adjusting

**Table 3. Summary of linear regression analysis demonstrating citizen's spirituality, health engagement, and attitudes with their acceptance and willingness to pay for the COVID-19 vaccine (n = 1423).**

| Variable | Acceptance | | Willingness to pay | |
|---|---|---|---|---|
| | Unadjusted coef. β (95% CI) | Adjusted coef. β (95% CI)[a] | Unadjusted coef. β (95% CI) | Adjusted coef. β (95% CI)[b] |
| Spirituality | 0.02 | 0.01 | -0.01 | -0.01 |
| | (0.02~0.03)** | (0.01~0.02)** | (-0.02~-0.01)* | (-0.02~-0.01)* |
| Health engagement (HE) | 0.16 | 0.09 | 0.07 | 0.03 |
| | (0.16~0.17)** | (0.08~0.10)** | (0.06~0.09)** | (0.01~0.05)* |
| Attitudes towards vaccines (AVs) | 0.39 | 0.18 | 0.18 | 0.04 |
| | (0.37~0.42)** | (0.16~0.21)** | (0.15~0.22)** | (0.01~0.09)* |
| Health beliefs–Perceived susceptibility (PSU) | 0.07 | 0.01 | 0.07 | 0.04 |
| | (0.06~0.08)** | (-0.01~0.02) | (0.06~0.09)** | (0.01~0.07)* |
| Health beliefs–Perceived severity (PSE) | 0.08 | 0.02 | 0.07 | -0.01 |
| | (0.06~0.09)** | (-0.01~0.03) | (0.05~0.08)** | (-0.04~0.02) |
| Health beliefs–Perceived benefits (PBE) | 0.10 | 0.03 | 0.10 | 0.07 |
| | (0.09~0.11)** | (0.02~0.04)** | (0.08~0.11)** | (0.05~0.08)** |
| Health beliefs–Perceived barriers (PBA) | -0.08 | -0.03 | -0.03 | -0.01 |
| | (-0.10~-0.07)** | (-0.04~-0.02)** | (-0.05~-0.01)* | (-0.03~0.01) |

AVs = attitude towards vaccines; β = beta; CIs = confidence intervals; COVID-19 = coronavirus disease 2019; HE = health engagement; PBA = perceived barriers; PBE = perceived benefits; PSE = perceived severity; PSU = perceived susceptibility. Adjusted beta-coefficients (coef.) and 95% CIs were estimated using a multiple linear regression after adjusting for [a] geographical region or [b] gender, age, income, geographical region, urbanicity, pandemic impact on income.

* $p<0.05$;
** $p<0.001$.

for confounding variables. Finally, S4 Table revealed that the six items of health beliefs constructs (PSU1, PSE3, PBE1, PBE3, PBA1, and PBA3) were the strongest determining factors of the vaccine acceptance, while other items of health beliefs (PSU2, PSU2, PSE1, PSE2, PBE2, and PBA2) were not significant predictors of the vaccine acceptance score after adjusting for confounding variables. Citizens who justified (chose 'agree') all items of HE and AV1 had no significant correlation with the willingness to pay score compared to those who responded with 'disagree' by adjustment for possible confounding factors. Citizens who agreed with the statement PSU1, PBE2, and PBA2 had significantly higher scores for willingness to pay, and they were sure of the vaccine's effectiveness in preventing infectious diseases (AVs2) (β = 0.47, 95% CI = 0.25~0.68; β = 0.24, 95% CI = 0.05~0.42; β = 0.17, 95% CI = 0.01~0.34, and β = 0.17, 95% CI = 0.01~0.33, respectively). Moreover, citizens who also agreed with the statement that they could not be bothered to get a vaccination (PBA3) had significantly lower scores for willingness to pay (β = -0.44, 95% CI = -0.57~-0.32).

## Discussion

To the best of our knowledge, this is the first study to assess Indonesian acceptance and willingness to pay for a COVID-19 vaccine. Herein, Indonesian citizens who agreed with having higher spirituality were statistically related with higher scores of vaccine acceptance among the Indonesian population. However, higher spirituality was significantly related with higher score of willingness to pay for a COVID-19 vaccine after adjustment for possible confounding factors. Similarly, Thomas et al. [42] reported that spirituality was strongly associated with parents' perceptions of their influence over and ways of dealing with health problems potentially related to the human papillomavirus vaccination. Remarkably, the occurrence of challenges escalated by the COVID-19 disease outbreak demonstrates the magnitude of spirituality [7], but spirituality's effects on acceptance and willingness to pay for a vaccine have not exclusively been explored or investigated. Indonesia has a diverse culture and extremely unique spiritual beliefs. This rare situation requires a holistic care of nursing including spiritual needs and how those needs are related to health behaviors and health beliefs [7, 23]. Conceivably, spirituality encourages acceptance-based responses, specifically, adaptive responses involving (a) being aware of and accepting of one's own emotional experiences, (b) learning a variety of coping mechanisms so that one can respond flexibly and interactively to emotional experiences while remaining committed to achieving recovery-related priorities, (c) implementing adaptive mechanisms as these states appear, and showing great outcomes as a result of those actions [43]. Our current findings indicated that spirituality might be strongly correlated with a willingness to pay and vaccine acceptance. Consequently, well-designed strategies to prevent or grow spirituality may even be important to increase the desire to be vaccinated.

It was found that HE was significantly related with acceptance vaccination, and it was a good predictor of individuals maintaining and improving a good attitude toward the vaccine [24]. Notably, those findings aligned with our results where HE and AVs indicated higher vaccine acceptance. A similar study, suggested that a comprehensive understanding of student' viewpoints on supporting their HE and consciousness may enable planning of effective responses and multidisciplinary educational strategies, including underlying AVs that influence perspectives about acceptance of the vaccine [44]. Therefore, these data suggested that the HE and AVs are predictive of COVID-19 vaccination acceptance. Currently, no studies have explored relationships between HE and a willing to pay for a vaccination. Of note, HE was a critical predictor of preventive behaviors [45]. In our present findings, positive AVs1 was not correlated with a willingness to pay for a vaccine, but AVs2 was related with a willingness to pay for a vaccine.

Our results were aligned with a previous study in terms of identifying that the HBM-PSU was related to willing to pay [9]. In particular, our findings also evaluated the PSE, but only one question was related to being afraid of getting COVID-19, and it was significantly associated with a high vaccine acceptance score but not significantly with a willingness to pay for the vaccination. Similar HBM outcomes, specifically PSE, were identified in a Malaysian population. Additionally, they showed that the public strong reliance of the HBM-PSE was correlated with a willingness to pay for a vaccine, but it was not positively correlated with vaccine acceptance [9]. However, a large community study of 1200 citizens in Hong Kong revealed that the PSE had a 1.16-fold higher score of vaccine acceptance after adjusting for covariates [33]. These conflicting results may be attributable to monthly income [9] and low levels of knowledge about vaccine programs [18]. In our data, high-scores on PBE, i.e., PBE1 and PBE3," were strongly determinants of vaccine acceptance after adjusting for covariates. Nevertheless, citizens with high scores on PBE of the immunization could decline their chances of getting COVID-19. This survey indicated that in terms of benefits, citizens who intended to receive the vaccine saw extraordinarily strong PBE in getting the vaccination to ensure their own and others' safety, linier to what Shmueli et al. suggested that vaccination enforcement relies on personal risk-benefit perceptions [31]. Similar to our study, a cross-sectional study in Kenya revealed that perceptions of vaccine benefits were associated with a willingness to pay for a Peste des Pettis Ruminants vaccine [46]. Citizens with health beliefs about side-effects of the vaccine interfering with their daily activities and those who could not be bothered to get an immunization were correlated with vaccine acceptance, but only one item of PBA "cannot be bothered to get an immunization" was correlated with willingness to pay. The present study aligned with previous investigations which suggested that citizens with a lower score for worrying about possible side-effects of the vaccination had a 1.81-fold lower score for vaccine acceptance, and no significant correlation with a willingness to pay for a COVID-19 vaccination [9]. Moreover, a high score on the PBA of "cannot be bothered to get the vaccine" was a significant predictor of a lower vaccine acceptance score among 799 general citizens in the US [32]. The increase in COVID-19-related skepticism of vaccine acceptance and the low rate of willingness to pay for a vaccine among Indonesian residents require further priority advocacy of health belief construct prevention, in which the HBM is obligatory among individual with skepticism of vaccine acceptance and with a low rate of willingness to pay for a vaccine. Based on the previous studies and considering skepticism over vaccine acceptance and the low rate of willingness to pay for a COVID-19 vaccine, delivering citizens with accurate health knowledge is the practical way to prevent such problems. Governments must ascertain and propagate proper COVID-19 vaccine-related information [33, 47]. Also, considering health beliefs with health education programs could be more serviceable and might be used to construct an HBM intervention [9, 33].

## Limitations of this study

The present study is not without limitations. First, the online evaluation methodology experienced a selection bias because only information on the Google form was shared via WhatsApp, Facebook, Instagram, Telegram, and Twitter. Since many people (approximately 61.8%) rely on technology to access online social media services [48], there was a risk that those (38.2%) who do not use media technology would be unable to access this form. In this study, we didn't test different price ranges for comparison and another limitation was a lack of citizens' prevalence from the eastern and central region and an International Standard Classification of Education of <3 education level, as this may implicate the generalizability of the findings and

which future research might specifically seek to enroll. However, we adjusted for a considerable number of potential confounding factors to be obtained by performing a multiple linear regression, thus minimizing the effect of an unequal distribution.

## Conclusion

Finally, the HE, AVs, and HBM were positively determinant factors of the intention to get vaccinated and willingness to pay for a vaccine. Our key findings show that spirituality was independently correlated with potential vaccine acceptance and willingness to pay for a vaccine. The willingness to pay and intention to get vaccinated could be impaired by worries regarding the side-effects of a vaccination interfering with daily activity of citizens. These constructs and independent predictors that were established include an implementation of vaccination strategies which really aim to escalate intention to accept vaccinated and willing to pay for it. Our findings offer to health professionals including nursing identifying and incorporating clinical counseling interventions strengthening HE, AVs, HBM, and spirituality to successfully boost the acceptance and willingness to pay. Furthermore, it provided government policy-making to boost citizen's immunization programs. The data gathered from this survey would provide scientific evidence for developing targeted programs to improve acceptance and willingness to pay for vaccines and enhance vaccine management strategic decisions for current and future.

## Supporting information

**S1 Checklist. The strengthening the reporting of observational studies in epidemiology (STROBE) protocol.**
(DOCX)

**S1 Data. Characteristics and acceptance and willingness data.**
(XLSX)

**S1 Table. Comparisons of citizen's spirituality, health engagement, and attitudes with their acceptance and willingness to pay for the COVID-19 vaccine ($n$ = 1423).**
(DOCX)

**S2 Table. Comparisons of citizen's health beliefs with their acceptance and willingness to pay for COVID-19 vaccine ($n$ = 1423).**
(DOCX)

**S3 Table. Adjusted beta-coefficients and 95% confidence intervals (CIs) of spirituality, health engagement, and attitude toward vaccine with participants' acceptance and willingness to pay for COVID-19 vaccine ($n$ = 1423).**
(DOCX)

**S4 Table. Adjusted beta-coefficients and 95% confidence intervals (CIs) of health beliefs constructs with participants' acceptance and willingness to pay for COVID-19 vaccine ($n$ = 1423).**
(DOCX)

**S1 Data Set.**
(DOC)

**S1 File.**
(DOC)

## Acknowledgments

The authors express their appreciation to the anonymous reviewers for their insightful feedback. We appreciate the time and effort of the survey respondents who participated voluntarily and made this research possible in the midst of the pandemic and strict lockdowns.

## Author Contributions

**Conceptualization:** Sri Handayani, Yohanes Andy Rias, Maria Dyah Kurniasari, Ratna Agustin, Yafi Sabila Rosyad, Ya Wen Shih, Hsiu Ting Tsai.

**Data curation:** Sri Handayani, Yohanes Andy Rias, Ya Wen Shih, Hsiu Ting Tsai.

**Formal analysis:** Sri Handayani, Yohanes Andy Rias, Ya Wen Shih.

**Funding acquisition:** Hsiu Ting Tsai.

**Investigation:** Sri Handayani, Yohanes Andy Rias, Maria Dyah Kurniasari, Ratna Agustin, Yafi Sabila Rosyad.

**Methodology:** Sri Handayani, Yohanes Andy Rias, Maria Dyah Kurniasari, Ching Wen Chang, Hsiu Ting Tsai.

**Project administration:** Sri Handayani, Yohanes Andy Rias.

**Supervision:** Ching Wen Chang, Hsiu Ting Tsai.

**Validation:** Sri Handayani, Yohanes Andy Rias, Maria Dyah Kurniasari, Ratna Agustin, Yafi Sabila Rosyad, Ya Wen Shih, Ching Wen Chang, Hsiu Ting Tsai.

**Visualization:** Sri Handayani, Yohanes Andy Rias.

**Writing – original draft:** Sri Handayani, Yohanes Andy Rias, Maria Dyah Kurniasari, Ya Wen Shih.

**Writing – review & editing:** Sri Handayani, Yohanes Andy Rias, Maria Dyah Kurniasari, Ratna Agustin, Yafi Sabila Rosyad, Ya Wen Shih, Ching Wen Chang, Hsiu Ting Tsai.

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
