## [Decision Letter · Decision Letter 0]

21 Jun 2022

PONE-D-22-14548Effects of spirituality, health engagement, health belief and attitudes toward acceptance and willingness to pay for a COVID-19 vaccinePLOS ONE

Dear Dr. Tsai,

Thank you for submitting your manuscript to PLOS ONE. After careful consideration, we feel that it has merit but does not fully meet PLOS ONE’s publication criteria as it currently stands. Therefore, we invite you to submit a revised version of the manuscript that addresses the points raised during the review process.

 Please submit your revised manuscript by Aug 05 2022 11:59PM. If you will need more time than this to complete your revisions, please reply to this message or contact the journal office at plosone@plos.org. Please include the following items when submitting your revised manuscript:A rebuttal letter that responds to each point raised by the academic editor and reviewer(s). You should upload this letter as a separate file labeled 'Response to Reviewers'.A marked-up copy of your manuscript that highlights changes made to the original version. You should upload this as a separate file labeled 'Revised Manuscript with Track Changes'.An unmarked version of your revised paper without tracked changes. You should upload this as a separate file labeled 'Manuscript'.

We look forward to receiving your revised manuscript.

Kind regards,

Harapan Harapan, MD, PhD

Academic Editor

PLOS ONE

Journal Requirements:

"Funding: This research was funded by the Ministry of Science and Technology

(MOST), Taiwan, through grant nos. 106-2314-B-038-013-MY3 and 109-2314-B-038-110-

MY3. "

"The funders had no role in study design, data collection and analysis, decision to publish, or preparation of the manuscript.

Funding: This research was funded by the Ministry of Science and Technology (MOST), Taiwan, through grant nos. 106-2314-B-038-013-MY3 and 109-2314-B-038-110-MY3 by H.T.T"

Reviewers' comments:

Reviewer's Responses to Questions

**Comments to the Author**

1. Is the manuscript technically sound, and do the data support the conclusions?

Reviewer #1: Partly

Reviewer #2: No

Reviewer #3: Partly

2. Has the statistical analysis been performed appropriately and rigorously? 

Reviewer #1: Yes

Reviewer #2: No

Reviewer #3: No

3. Have the authors made all data underlying the findings in their manuscript fully available?

Reviewer #1: Yes

Reviewer #2: No

Reviewer #3: No

4. Is the manuscript presented in an intelligible fashion and written in standard English?

Reviewer #1: No

Reviewer #2: No

Reviewer #3: No

5. Review Comments to the Author

Reviewer #1: Dear authors,

I read your manuscript titled: Effects of spirituality, health engagement, health belief and attitudes toward

acceptance and willingness to pay for a COVID-19 vaccine, with interest.

Here are some revisions required from your side.

1. Please carry out an extensive English language editing.

2. The methodology, already compromised using a convenient sampling technique, needs further justification as to how the single responder did not use multiple social media platforms to respond to your questionnaire.

3. There are numerous limitations to your study, especially with regards to the generalisability of the findings and also the validity of the tools used for assessment.

4. Please add a section on the validity of the assessment tool utilised.

5. Please be consistent in using the terms; multivariate and multiple regression, both have different understanding.

6. Please avoid using abbreviations in the abstract.

7. Please elaborate on the implications of the findings from your investigation.

Reviewer #2: 1. In the abstract. All abbreviations should be defined.

2. If supported by the journal format. I recommend the authors to make a list of abbreviations.

3. The English should be checked and revised. The use of decimal separator should be taken carefully, such as “50.404 having died”.

4. The logical flow is confusing, for instance paragraph 1 and 2 in the introduction.

5. Statement “Vaccine acceptance and willingness to pay..” needs citation. I recommend: Sallam et al. Narra J 2022; 2(1): e74 – doi: 10.52225/narra.v2i1.74

6. After this full sentence, “Recent studies shown that the acceptance of vaccination…” I recommend to include this study because it longitudinally compare data from multiple countries Rosiello et al. Narra J 2021; 1(3): e55-doi: 10.52225/narra.v1i3.55

7. Introduction is too long. Many redundant paragraph, I recommend to trim some of them.

8. “18 provinces in Indonesia” out of how many provinces?

9. “the most accessible online media networks used by Indonesian citizens” Needs citation or removed.

10. “Indonesian citizens” What parameters determine the participants are Indonesian citizens. Do you have any specific inclusion/exclusion criteria for the citizenship?

11. “1,423 samples” how this number is determined?

12. Include the p value when describing the results. Add 0 before (.) in decimal.

13. In the table footnote, authors indicate * for statistical significant at p<0.05 and ** -- at p<0.001. But no asterisk was put on the table data.

14. The data are too many and confusing. Please only include significant data on the paper. The rest can be put in Supplementary file. Regardless, this is just a suggestion.

15. Again. The discussion has rather confusing logical flows. For example, the authors highlight the significance of their work at the beginning of the subsection. This leads the explanations to obtain inadequate comparison. In addition, authors may divide the discussion into several subsections.

16. “Indonesia has a diverse culture and extremely unique spiritual belief..” Is it possible the data could be biased because of such extreme heterogeneity? If yes, how did author overcome this?

17. Where are the underlying data? Otherwise the reasons are stated, the journal requires the publication of underlying data.

Reviewer #3: • The title should be changed since this study was a cross sectional study therefore it is not possible to measure the impact or effect and it is advised to change the titled to the relationship rather than measuring the effect

• Research objectives are not clearly stated in the introduction section and that way it is advised to revise and mention about related research objective

• it is required to provide scientifical calculation for sample size for the study

• data collection procedures should be explained in details and it is not clear what does it mean researchers technological and personal networks please elaborate more on this section

• Most of explanation on their data collection procedures is related to explaining different section of the questionnaire rather than the process of data collection

• It is not mention about the process of translation since the original question are were in English the process of cross cultural adaptation should be explained in details like forward backward translation cognitive debriefing

• It is required to mention about the scoring of all instruments

• Results of face and content validity should be elaborated by using content validity index and Kappa

• For the reliability analysis reporting only Alpha Cronbach is not adequate and it is advice also to mention about the total item correlation for each indicator

• What is “AVs uses “ in page 9

• Since in this study parametric tests were applied, it is required to mention about the normality test of distribution for all research variables

• The assumption of the homogeneity of variance for ANOVA also need to be reported

• In Table 1 comparison between geographical region was done using one way ANOVA since the sample size or not equal for western eastern and central regions therefore it is advised to use Kruskal Wallis test rather than one way ANOVA

• Since this study used and non-random sampling therefore the P value for interpretation of the results is not applicable therefore it is advised to discuss and interpret defining based on the effect size rather than P value

• In Table 2 and 3 comparing between disagree/ agree for each indicator was done, which is not required to do the comparison based on indicators. it is advised to concentrate on the overall score of a scale and its association with willingness and also acceptance

• Multiple linear regression analysis should be based on the total score of the components rather than including all indicators in the questionnaire as predictors therefore it is recommended to revise and redo the analysis for multiple linear regression based on the total score of all predictors in one model

6. PLOS authors have the option to publish the peer review history of their article (what does this mean?). If published, this will include your full peer review and any attached files.

Reviewer #1: No

Reviewer #2: **Yes: **Muhammad Iqhrammullah

Reviewer #3: No

---

## [Author Response · Author response to Decision Letter 0]

12 Aug 2022

RE: [PONE-D-22-14548] - [EMID: ccb1e548dd430d0c]-Version 1

Response to Reviewer 1 Comments

Dear Reviewer #1,

Thank you for considering our manuscript and for the valuable suggestions, also the opportunity to resubmit a revised manuscript, which helps us to improve the article. We carefully revised the manuscript in accordance with your comments. The revised sections of the manuscript are marked with red color. Our point-by-point responses to the comments are as follows. We very much hope the revised manuscript is accepted for publication in PLOS ONE. Thank you very much for your consideration.

Point 1. Please carry out an extensive English language editing.

Response 1: Thank you for your valuable suggestion. This revised manuscript was edited by Taipei Medical University Academic Editing.

Point 2. The methodology, already compromised using a convenient sampling technique, needs further justification as to how the single responder did not use multiple social media platforms to respond to your questionnaire.

Response 2: Thank you for your comments. With the issue of duplicate response, we used participants email to avoid overlapping response during data collection (Please see line 139–140 on page 6)

Point 3. There are numerous limitations to your study, especially with regards to the generalizability of the findings and also the validity of the tools used for assessment.

Response 3: Thank you for your comments. We add several new information to clarify and revise this point to make it clearer and more precise based on the reviewer’s suggestion as follows:

Generalizability:

“The sample size was calculated based on estimates from the distribution of the general population as reported by the Central bureau of statistics, Indonesia. Proportions from eastern, central and western regions of Indonesia are reported at 2.76%, 16,14% and 81.10% respectively [38]. In our study, we reached participants from all regions of Indonesia and obtained 11.9%, 16.9% and 71.2% from each base, which has a similar pattern to the proportional distribution of these regions in the general population” (Please see line 240–245 on page 10).

“Another limitation was a lack of citizens’ prevalence from the eastern and central region and an International Standard Classification of Education of <3 education level, as this may implicate the generalizability of the findings and which future research might specifically seek to enroll. However, we adjusted for a considerable number of potential confounding factors to be obtained by performing a multiple linear regression, thus minimizing the effect of an unequal distribution” (Please see line 428–433 on page 19).

Validity of the tools used for assessment

In our manuscripts we already mention content validity index (CVI) and kappa (k*). Moreover, we add new result of The Kaiser–Meyer–Olkin (KMO), kappa, Bartlett’s tests of sphericity, a Cronbach's alpha and item-total correlation analysis were used to determine validity and reliability of the tools used for assessment (Please see line 191–194 on page 8).

Point 4. Please add a section on the validity of the assessment tool utilized.

Response 4: Thank you for your valuable comment and suggestion. In this revised manuscript, we added a description to make clear the validity of the assessment tool utilized based on the reviewer’s suggestion as follows in the section of the methods of our study.

“Further, we reviewed cognitive debriefing results and the finalized version with content validity index (CVI) and kappa (k*). Finally, we conducted an analysis of the reliability and validity with the Kaiser-Meyer-Olkin (KMO) test, the Bartlett’s test of sphericity value, Cronbach’s alpha and item-total correlation coefficient” (Please see line 191–194 on page 8).

“In our study, HE questionnaire English was translated into Indonesian and had a CVI of 0.93, k* of 0.94 to 1, the value of the KMO test was 0.72 and the Bartlett’s test of sphericity value was significant (p < 0,001). Furthermore, Cronbach's alpha of 0.91 with item-total correlation coefficient score was 0.68 to 0.88” (Please see line 200–204 on page 8).

“The Indonesian version of the VAs questionnaire had an acceptable CVI 0.95 with k* of 0.98 to 1. The value of the KMO test was 0.69 and the Bartlett’s test of sphericity value was significant (p < 0,001). Furthermore, a total Cronbach's alpha of 0.70 with item-total correlation coefficient score was 0.60 and 0.68 in our study” (Please see line 212–216 on page 9).

“In our study, the questionnaire of the HBM Indonesian version presented that the CVI was 0.95 with k* of 0.89 to 0.92. The value of the KMO test was 0.61 and the Bartlett’s test of sphericity value was significant (p < 0,001). Furthermore, a total of Cronbach's alpha of 0.81 with item-total correlation coefficient score was 0.63 to 0.71” (Please see line 225–229 on page 9-10).

Point 5. Please be consistent in using the terms; multivariate and multiple regression, both have different understanding.

Response 5: Thank you for your valuable comment and suggestion. We revised “multivariate regression” to “multiple regression” (Please see abstract, line 38 on page 2).

Point 6. Please avoid using abbreviations in the abstract.

Response 6: Thank you for your valuable comment and suggestion. We appreciate this reviewer’s comment. In this revised manuscript, we avoid using abbreviations in the conclusions of abstract section.

“Conclusions: Results demonstrated the utility of spirituality, health engagement, health belief model, and attitudes towards vaccines in understanding acceptance and willingness to pay for a vaccine. Specifically, a key obstacle to the acceptance of and willingness to pay COVID-19 vaccination included a high score of the perceived barrier construct. Moreover, the acceptance of and willingness to pay could be impaired by worries about the side-effects of a COVID-19 vaccination” (Please see abstract, line 45–50 on page 2).

Point 7. Please elaborate on the implications of the findings from your investigation.

Response 7: Thank you very much. We appreciate this reviewer’s comments. In this revised manuscript, we added a description about implications of this study as follows in the section of the conclusion

“Our findings offer to health professionals including nursing identifying and incorporating clinical counseling interventions strengthening HE, AVs, HBM, and spirituality to successfully boost the acceptance and willingness to pay. Furthermore, it provided to government policy-making to boost citizen's immunization programs. The data gathered from this survey would provide scientific evidence for developing targeted programs to improve acceptance and willingness to pay vaccine and enhance vaccine management strategic decisions for current and future” (Please see abstract, line 443–449 on page 20).

thank you---------------------------------------------------

Response to Reviewer 2 Comments

Dear Reviewer #2,

Thank you for considering our manuscript and for the valuable suggestions, also the opportunity to resubmit a revised manuscript, which helps us to improve the article. We carefully revised the manuscript in accordance with your comments. The revised sections of the manuscript are marked with red color. Our point-by-point responses to the comments are as follows. We very much hope the revised manuscript is accepted for publication in PLOS ONE. Thank you very much for your consideration.

Point 1. In the abstract. All abbreviations should be defined.

Response 1: Thank you very much. We appreciate your comment. In this revised manuscript, we avoid using abbreviations based on reviewer’s comment and provide full name of the abbreviations in the abstract section as follows;

“Conclusions: Results demonstrated the utility of spirituality, health engagement, health belief model, and attitudes towards vaccines in understanding acceptance and willingness to pay for a vaccine. Specifically, a key obstacle to the acceptance of and willingness to pay COVID-19 vaccination included a high score of the perceived barrier construct. Moreover, the acceptance of and willingness to pay could be impaired by worries about the side-effects of a COVID-19 vaccination” (Please see abstract, line 44–49 on page 2).

Point 2. If supported by the journal format. I recommend the authors to make a list of abbreviations.

Response 2: Thank you for your valuable comment and suggestion. All abbreviations have been explained at the beginning of the previous sentences follow the submission guidelines as follows;

Abbreviations: Adjusted beta coefficients (�); Content validity index (CVI); Confidence intervals (CIs); Coronavirus disease 2019 (COVID-19); Daily spiritual experiences scale (DSES); Exploratory factors analysis (EFA), Health belief model (HBM); Health engagement (HE); IDR = Indonesian rupiah; ISCED = International standard classification of education; Kaiser-Meyer-Olkin (KMO); One-way analysis of variance (ANOVA); Perceived barriers (PBA), Perceived benefits (PBE); Perceived severity (PSE); Perceived susceptibility (PSU), Standard deviation (SD), Vaccine attitudes (AVs); Variance inflation factor (VIF); World Health Organization (WHO).

Point 3. The English should be checked and revised. The use of decimal separator should be taken carefully, such as “50.404 having died”.

Response 3: Thank you for your valuable comments and suggestions. In order to make data better presented, we reorganized the sentences based on the reviewer’s suggestion.

“Additionally, this disease has spread to Indonesia, where approximately 1,816,041 people are reported to be infected, with 50,404 deaths” (Please see line 61–63 on page 3)

Point 4. The logical flow is confusing, for instance paragraph 1 and 2 in the introduction.

Response 4: Thank you for your valuable comment. In order to make manuscripts to be better presented with precise and logical flow, we re-organize the sentences (paragraph 1 and 2) based on the reviewer’s suggestion as follows;

 “COVID-19 caused clusters of a complex respiratory syndrome characterized with a novel beta-coronaviruses infection [1]. As of May 31, 2021, the WHO confirmed that 170,051,718 individuals had been infected with COVID-19 worldwide [2]. Additionally, this disease has spread to Indonesia, where approximately 1,816,041 people are reported to be infected, with 50,404 deaths [3]. After scientists discovered this new SARS-CoV-2 strain, vaccines for COVID-19 were rapidly developed to be distributed globally [4, 5]. While vaccine programs could substantially alleviate the spread of the virus, one of the problems for policymakers is determining how to motivate their citizens to get vaccinated. Most vaccine skeptics refuse to be vaccinated [6]. Interestingly, Indonesia is unique because citizens typically have extremely spiritual beliefs, health attitude issues [7], and differences in health perspective [8], which may influence acceptance and willingness to pay COVID-19 vaccine.” (Please see paragraph 1 in the introduction, line 59–69 on page 3)

Point 5. Statement “Vaccine acceptance and willingness to pay..” needs citation. I recommend: Sallam et al. Narra J 2022; 2(1): e74 – doi: 10.52225/narra.v2i1.74

Response 5: Thank you for your valuable comment and suggestion. In this revised manuscript, we added a new reference number10 based on reviewer’s suggestion as follows;

“Acceptance and willingness to pay for a COVID-19 vaccine are critical to the success of a high-coverage vaccination program [9, 10]” (Please see line 71 on page 3)

Point 6. After this full sentence, “Recent studies shown that the acceptance of vaccination…” I recommend to include this study because it longitudinally compare data from multiple countries Rosiello et al. Narra J 2021; 1(3): e55-doi: 10.52225/narra.v1i3.55

Response 6: Thank you for your valuable comment and suggestion. In this revised manuscript, we added an information and new reference based on reviewer’s suggestion as follows;

“Moreover, an epidemiological study in low- or middle-income countries such as Bangladesh, India, Iran, Pakistan, Egypt, Nigeria, Sudan, Tunisia, Brazil, and Chile presented that the acceptance of vaccination was approximately 58.3 % to 80.1%” (Please see line 73–76 on page 3)

Point 7. Introduction is too long. Many redundant paragraphs, I recommend to trim some of them.

Response 7: Thank you for your valuable comment. In order to make data better presented, we reorganized (which were marked with red color) and trimmed some descriptions based on the reviewer’s suggestion (Please see introduction section, line 59–118 on page 3–5).

Point 8. “18 provinces in Indonesia” out of how many provinces?

Response 8: Thank you for your valuable comment. In this revised manuscript, we added an information based on reviewer’s suggestion as follows;

“A cross-sectional online-based overview during COVID-19 for 18 provinces out of 34 provinces in Indonesia” (Please see line 122–123 on page 5)

Point 9. “the most accessible online media networks used by Indonesian citizens” Needs citation or removed.

Response 9: Thank you for your valuable comment and suggestion. We removed “the most accessible online media networks used by Indonesian citizens” based on the reviewer’s suggestion. 

Point 10. “Indonesian citizens” What parameters determine the participants are Indonesian citizens. Do you have any specific inclusion/exclusion criteria for the citizenship?

Response 10: Thank you for your valuable comment. In this revised manuscript, we added an information based on reviewer’s suggestion as follows;

Indonesian citizens parameters are the original Indonesians and foreign nationals who are legally recognized as Indonesian citizens currently live in Indonesia.

“The eligible target population was Indonesian citizens aged 17 until 65 years, who understood Bahasa Indonesia, currently stay in Indonesia, and filled the consent form. Citizens who had previously been confirmed with suspected COVID-19 was excluded” (Please see line 124–126 on page 5)

Point 11. “1,423 samples” how this number is determined?

Response 11: Thank you for your valuable comment. In this revised manuscript, we added a sample size calculation based on reviewer’s suggestion as follows;

“Sample size was estimated based on previous study [36] with the formula; n = uap (1 − p)/δ2, where n = minimum desired sample size, ua = the standard normal deviation, usually set as 1.96 which corresponds to 5% level of significance. p = the average rate of acceptance of vaccine was estimated on the basis of the available literature and its value was set at 85% [37], δ = of precision set at 0.015. The calculated minimum sample size was 1,111 (n = 1.96 x 0.85 x (1- 0.85)/0.0152 = 1,111). We expected a potential missing data of 20% with a large population and thus aimed to recruit at least 1,388 participants. Finally, during one-month data collection, the total sample consisted of 1,423 Indonesian citizens” (Please see line 232–239 on page 10).

Point 12. Include the p value when describing the results. Add 0 before (.) in decimal

Response 12: Thank you for your valuable comment and suggestion. In order to make data to be better presented, we add “0” before “.” in decimal (results p value) based-on the reviewer’s suggestion (Please see Table 1, line 276–78 on page 11-12; Table 2, line 290-292 on page 13, Table 3, line 314 on page 14).

Point 13. In the table footnote, authors indicate * for statistically significant at p<0.05 and ** -- at p<0.001. But no asterisk was put on the table data.

Response 13: Thank you for your valuable comment. In this revised manuscript, we deleted “* p<.05; ** p<.001” in Table 1 footnote (Please see the footnote of Table 1; line 280 on page 12).

Point 14. The data are too many and confusing. Please only include significant data on the paper. The rest can be put in Supplementary file. Regardless, this is just a suggestion.

Response 14: Thank you for your valuable comment and suggestion. In order to make manuscripts to be better presented, we organize all data results and several data put in supplementary file based on the reviewer’s suggestion (Please see results sections; line 271–347 on page 11–16).

Point 15. Again. The discussion has rather confusing logical flows. For example, the authors highlight the significance of their work at the beginning of the subsection. This leads the explanations to obtain inadequate comparison. In addition, authors may divide the discussion into several subsections.

Response 15: Thank you for your valuable comment. In order to make manuscripts to be better presented with precise and logical flow, we reorganize and divide the discussion into several subsections based on the reviewer’s suggestion (Please see line 350–420 on page 16–19).

Point 16. “Indonesia has a diverse culture and extremely unique spiritual belief….” Is it possible the data could be biased because of such extreme heterogeneity? If yes, how did author overcome this? Where are the underlying data? Otherwise the reasons are stated, the journal requires the publication of underlying data.

Response 16: We appreciate your insightful comments. The Daily Spiritual Experiences Scale (DSES) questionnaire is often used in epidemiological research, and people of different religions, cultures, and traditions have been suggested as a reason why.

The DSES instrument (Underwood and Teresi 2002; Underwood 2006) was designed on the basis of extensive research involving analysis of sources from theology, religion, and social sciences, investigation of spirituality measurements, in-depth interviews and focus groups with people from different religions, cultures, traditions. The DSES instrument was developed to assess the daily frequency of specific experiences of spirituality and interaction with transcendence. Items are designed to measure spiritual experience, not beliefs or behavior based on religious and spiritual doctrines. Spiritual experiences may be evoked by a religious context or by daily events, individual religion or religious or spiritual beliefs. Moreover, The DSES is composed of various concepts: transcendent connection, the support provided by God, divine or transcendent, inner peace and harmony, interconnectedness with all living things, reverence for beauty, gratitude, compassion, mercy, and the desire to be closer to God.

The tool is validated in many languages, widely used and applicable to people with different religious traditions or atheists or agnostics (Underwood 2006; Ellison and Fan 2008; Kalkstein and Tower 2009; Ng et al. 2009; Bailly and Roussiau 2010; Sánchez et al. 2010; Underwood 2011; Loustalot et al. 2011; Kimura et al. 2012; Rakošec et al. 2015; Lo et al. 2016).

thank you---------------------------------------------------

Response to Reviewer 3 Comments

Dear Reviewer #3,

Thank you for considering our manuscript and for the valuable suggestions, also the opportunity to resubmit a revised manuscript, which helps us to improve the article. We carefully revised the manuscript in accordance with your comments. The revised sections of the manuscript are marked with red color. Our point-by-point responses to the comments are as follows. We very much hope the revised manuscript is accepted for publication in PLOS ONE. Thank you very much for your consideration.

Point 1. The title should be changed since this study was a cross sectional study therefore it is not possible to measure the impact or effect and it is advised to change the titled to the relationship rather than measuring the effect

Response 1: Thank you for your valuable comment and suggestion. We re-word “The effect” to “Relationship” based on the reviewer’s suggestion in the title section.

“Relationship of spirituality, health engagement, health belief and attitudes toward acceptance and willingness to pay for a COVID-19 vaccine”

Point 2. Research objectives are not clearly stated in the introduction section and that way it is advised to revise and mention about related research objective

Response 2: Thank you for your valuable comment and suggestion. In order to make manuscripts to be better presented with precise and logical flow, we organize the research objective based on the reviewer’s suggestion as follows; 

“To fill these gaps, this study explored how Indonesians accepted the COVID-19 vaccine and their willingness to pay for it. This was accomplished by surveying their spirituality, HE, HBM constructs, and AVs” (Please see line 116–118 on page 5).

Point 3. it is required to provide scientifically calculation for sample size for the study

Response 3: Thank you for your valuable comment. In this revised manuscript, we added a sample size calculation based on reviewer’s suggestion as follows;

“Sample size was estimated based on previous study [36] with the formula; n = uap (1 − p)/δ2, where n = minimum desired sample size, ua = the standard normal deviation, usually set as 1.96 which corresponds to 5% level of significance. p = the average rate of acceptance of vaccine was estimated on the basis of the available literature and its value was set at 85% [37], δ = of precision set at 0.015. The calculated minimum sample size was 1,111 (n = 1.96 x 0.85 x (1- 0.85)/0.0152 = 1,111). We expected a potential missing data of 20% with a large population and thus aimed to recruit at least 1,388 participants. Finally, during one-month data collection, the total sample consisted of 1,423 Indonesian citizens.” (Please see line 232–239 on page 10).

Point 4. data collection procedures should be explained in details and it is not clear what does it mean researchers technological and personal networks please elaborate more on this section

Response 4: Thank you for your valuable comments and suggestions. In order to make manuscripts to be better presented with precise and detail, we reorganize the data collection procedures based on the reviewer’s suggestion as follows; 

“The online survey was distributed using a Google Form link that was shared on social media platforms including WhatsApp, Instagram, Telegram, and Facebook. Furthermore, this relies on researchers’ technical and personal networks and engaging with and distributing the survey through social media influencers and community leaders. Participants were selected for the study using a simplified snowball sampling technique, and they were asked to forward the invitation to their contacts; the estimated completion time for the survey was 15 minutes. We conducted different procedures to target as many respondents as possible from across the region during the December 15, 2020 to January 12, 2021 data collection period. Finally, 1,423 people responded to our Google form” (Please see line 131–140 on page 6)

“The Google Form link had four sections. (1) Before allowing participants to proceed to the survey questions, the first section informed them of the objective of the study and eligibility requirements. Furthermore, the informed consent was taken by checking the box "Agree," which was required to confirm that they understood the authorization information and met the inclusion and exclusion criteria. Additionally, participants decided to participate voluntarily and with the freedom to withdraw at any time; (2) Second section comprised questions correlated to sociodemographic; (3) Third section comprised questions that assessed the intention to accept being vaccinated and willingness to pay for vaccinated; (4) Fourth section contained 35 questions including HE, AVs, HBM, and spirituality questionnaire. Finally, a page at the end expressed our gratitude, and all individuals who completed the survey were encouraged to persuade new respondents from their contact lists to participate by forwarding the link to the online survey” (Please see line 141–152 on page 6)

Point 5. Most of explanation on their data collection procedures is related to explaining different section of the questionnaire rather than the process of data collection

Response 5: Thank you for your valuable comments and suggestions. In order to make manuscripts to be better presented with precise and detail, we reorganize the data collection procedures based on the reviewer’s suggestion as follows; 

“The online survey was distributed using a Google Form link that was shared on social media platforms including WhatsApp, Instagram, Telegram, and Facebook. Furthermore, this relies on researchers’ technical and personal networks and engaging with and distributing the survey through social media influencers and community leaders. Participants were selected for the study using a simplified snowball sampling technique, and they were asked to forward the invitation to their contacts; the estimated completion time for the survey was 15 minutes. We conducted different procedures to target as many respondents as possible from across the region during the December 15, 2020 to January 12, 2021 data collection period. Finally, 1,423 people responded to our Google form” (Please see line 131–140 on page 6)

“The Google Form link had four sections. (1) Before allowing participants to proceed to the survey questions, the first section informed them of the objective of the study and eligibility requirements. Furthermore, the informed consent was taken by checking the box "Agree," which was required to confirm that they understood the authorization information and met the inclusion and exclusion criteria. Additionally, participants decided to participate voluntarily and with the freedom to withdraw at any time; (2) Second section comprised questions correlated to sociodemographic; (3) Third section comprised questions that assessed the intention to accept being vaccinated and willingness to pay for vaccinated; (4) Fourth section contained 35 questions including HE, AVs, HBM, and spirituality questionnaire. Finally, a page at the end expressed our gratitude, and all individuals who completed the survey were encouraged to persuade new respondents from their contact lists to participate by forwarding the link to the online survey” (Please see line 141–152 on page 6)

Point 6. It is not mention about the process of translation since the original question are were in English the process of cross-cultural adaptation should be explained in details like forward backward translation cognitive debriefing

Response 6: Thank you for your valuable comment and suggestion. In this revised manuscript, we added a description to make a clear the process of cross-cultural adaptation based on the reviewer’s suggestion as follows; 

“In the present study, the questionnaires including HE, AVs, and HBM were assessed for the translation process. After obtaining approval from the original authors, the questionnaires (HE, AVs, and HBM) were independently translated into Indonesian using the forward and back-translation methods. The questionnaires were translated by five translators, a certified translator and four experts in nursing research in Indonesian universities, whose native language was Indonesian and who were bilingual and fluent in English. The translators were assessing the questionnaire items to be relevant to measure the HE, AVs, and HBM toward acceptance and willingness to pay a COVID-19 vaccination precisely for linguistic and conceptual equivalence. In brief, Indonesian-speaking academics were first contacted to review the translated version for grammatical accuracy and clarity. Thus, four independent bilingual translators completed the back translation of the Bahasa edition into English. In addition, the final Bahasa version was obtained by comparing the original questionnaire with its back translation. Translators were instructed to avoid metaphors, colloquial terminology, and hypothetical statements, and to use simple sentences. Initially, prior to completing the formal online survey, we conducted a pilot study with 60 residents in the close surroundings of the researchers to determine the questionnaire's readability and reliability”. Further, we reviewed cognitive debriefing results and the finalized version with content validity index (CVI) and kappa (k*). Finally, we conducted an analysis of the reliability and validity with Kaiser-Meyer-Olkin (KMO) test, the Bartlett’s test of sphericity value, a Cronbach's alpha and item-total correlation coefficient” (Please see line 175–194 on page 7–8).

Point 7. It is required to mention about the scoring of all instruments

Response 7: Thank you for your valuable comment and suggestion. In this revised manuscript, we mention the scoring of all instruments based on the reviewer’s suggestion.

“The total score ranges from 1 to 5, a higher score indicated a more-favorable attitude to acceptance a COVID-19 vaccine” (Please see line 161–162 on page 7).

“The total score ranges from 1 to 5, the higher the score an individual has, the greater their willingness to pay for a vaccine” (Please see line 166–167 on page 7).

“The total score ranges from 16 to 96, the greater the number of experiences points a person has, the greater their spirituality. Participants‘ overall spirituality was categorized, as high if the score was � 72, and low if the score was <72 [7]” (Please see line 172–174 on page 7).

“The total score ranges from 6 to 30, a greater value indicating greater HE [24]. Interestingly, we defined HE score with response as continuous data on five-point Likert scale; 1 (definitely disagree) to 5 (strongly agree). Also, we defined HE scores as categorical data for disagreement (definitely disagree/disagree/strongly disagree) and agreement (agree/strongly agree) presented in S1 Table” (Please see line 196–200 on page 8).

“The total score ranges from 2 to 10, a greater value indicating greater AVs. For our study analysis, we defined AVs score with response as continuous (total score). Also, we defined VAs scores involving the agreement (strongly agree/agree), and disagreement (neither agree nor disagree, strongly disagree/disagree) presented in S1 Table” (Please see line 206–210 on page 9).

“Response this statement was ranked on a 7-point Likert-scale; 1 (strongly disagree) to 7 (strongly agree) [35]. Also, HBM constructs were used in COVID-19 vaccinations previous research [9, 32]. The total score ranges from 12 to 84, a higher score indicates a good health belief, except for the PBA construct. In the present study, we defined HBM score with response as continuous data or total score in each construct. Moreover, the detailed HBM constructs score involve the agreement (somewhat agree/agree/strongly agree), and disagreement (somewhat disagree/disagree/ strongly disagree /neither agree nor disagree) presented in S2 Table” (Please see line 219–225 on page 9).

Point 8. Results of face and content validity should be elaborated by using content validity index and Kappa

Response 8: Thank you for your valuable comment and suggestion. We added a kappa results to make a clear the results of face and content validity based on the reviewer’s suggestion as follows;

“HE questionnaire English was translated into Indonesian and had a CVI of 0.93, k* of 0.94 to 1” (Please see line 200-201 on page 8).

“The Indonesian version of the VAs questionnaire had an acceptable CVI 0.95 with k* of 0.98 to 1” (Please see line 212-213 on page 9).

“The questionnaire of the HBM Indonesian version presented that the CVI was 0.95 with k* of 0.89 to 0.92” (Please see line 226–227 on page 9).

Point 9. For the reliability analysis reporting only Alpha Cronbach is not adequate and it is advice also to mention about the total item correlation for each indicator

Response 9: Thank you for your valuable comment and suggestion. In this revised manuscript, we added a description to make a clear the reliability analysis report based on the reviewer’s suggestion as follows; 

HE questionnaire: “the value of the KMO test was 0.72 and the Bartlett’s test of sphericity value was significant (p < 0,001). Furthermore, a Cronbach's alpha of 0.91 with item-total correlation coefficient score was 0.68 to 0.88” (Please see line 201–204 on page 8).

AVs questionnaire: “The value of KMO test was 0.59 and the Bartlett’s test of sphericity value was significant (p < 0,001). Furthermore, a total Cronbach's alpha of 0.70 with item-total correlation coefficient score was 0.60 and 0.68 in our study” (Please see line 213–216 on page 9).

HBM questionnaire: “The value of KMO test was 0.61 and the Bartlett’s test of sphericity value was significant (p < 0,001). Furthermore, the total Cronbach's alpha of 0.81 with item-total correlation coefficient score was 0.63 to 0.71” (Please see line 227–229 on page 9).

Point 10. What is “AVs uses “in page 9

Response 10: Thank you for your valuable comment. We revised “AVs uses” to “Vaccine attitudes (AVs) consists of…” (Please see line 205 on page 9).

Point 11. Since in this study parametric tests were applied, it is required to mention about the normality test of distribution for all research variables. 

Response 11: Thank you for your valuable comment. In this revised manuscript, we added the normality test of distribution for all research variables through skewness and kurtosis test based on the reviewer’s suggestion as follows; 

 “Absolute values for skewness and kurtosis were used to assess normality of the data; skewness value of -0.264 and kurtosis value of 1.677 indicated a normal distribution [40]” (Please see line 254–256 on page 10-11).

References: Kim H-Y. Statistical notes for clinical researchers: assessing normal distribution (2) using skewness and kurtosis. Restorative dentistry & endodontics. 2013;38(1):52-4. https://doi.org/10.5395/rde.2013.38.1.52

Point 12. The assumption of the homogeneity of variance for ANOVA also need to be reported

Response 12: Thank you for your valuable comment and suggestion. We used the assumption of the homogeneity regarding previous references from Kim, 2013.

“For sample sizes greater than 300, depend on the histograms and the absolute values of skewness and kurtosis without considering z-values. Either an absolute skewness value larger than 2 or an absolute kurtosis (proper) larger than 7 may be used as reference values for determining substantial non-normality”

References: Kim H-Y. Statistical notes for clinical researchers: assessing normal distribution (2) using skewness and kurtosis. Restorative dentistry & endodontics. 2013;38(1):52-4. https://doi.org/10.5395/rde.2013.38.1.52

Point 13. In Table 1 comparison between geographical region was done using one-way ANOVA since the sample size or not equal for western eastern and central regions therefore it is advised to use Kruskal Wallis test rather than one-way ANOVA

Response 13: Thank you for your valuable comment and suggestion. In this revised manuscript, we added the normality test of distribution based on the reviewer’s suggestion as follows; 

“Absolute values for skewness and kurtosis were used to assess normality of the data; skewness value of -0.264 and kurtosis value of 1.677 indicated a normal distribution [40]” (Please see line 254–256 on page 10-11). Thus, we use one-way ANOVA in Table 1 (comparison between geographical region).

References: Kim H. Y. (2013). Statistical notes for clinical researchers: assessing normal distribution (2) using skewness and kurtosis. Restorative dentistry & endodontics, 38(1), 52–54. https://doi.org/10.5395/rde.2013.38.1.52

Point 14. Since this study used and non-random sampling therefore the P value for interpretation of the results is not applicable therefore it is advised to discuss and interpret defining based on the effect size rather than P value

Response 14: Thank you for your valuable comment and suggestion. We deleted the P value based on the reviewer’s suggestion. 

We presented the adjusted beta coefficients (�) with 95% confidence intervals (CIs) to interpret defining based on the effect size rather than P value (Please see results section table 3 explanation; line 306 –309 on page 14).

Point 15. In Table 2 and 3 comparing between disagree/ agree for each indicator was done, which is not required to do the comparison based on indicators. it is advised to concentrate on the overall score of a scale and its association with willingness and also acceptance

Response 15: Thank you for your valuable comments and suggestions. In order to make manuscripts better presented, we reorganize data Tables 2 and 3 become Table 2 with the overall score of a scale. Tables 4 and 5 become Table 3. However, we presented all indicator's data in a supplementary file based on the reviewer’s suggestion (Please see results sections; line 350–420on page 16–19).

Point 16. Multiple linear regression analysis should be based on the total score of the components rather than including all indicators in the questionnaire as predictors therefore it is recommended to revise and redo the analysis for multiple linear regression based on the total score of all predictors in one model

Response 16: Thank you for your valuable comments and suggestions. In order to make manuscripts better presented, we reorganize data and redo the analysis. Data in tables 4 and 5 become table 3 with the overall score of a scale. However, we presented all the indicator's data in a supplementary file based on the reviewer’s suggestion Please see results section table 3 explanation; line 306 –309 on page 14).

thank you---------------------------------------------------

---

## [Decision Letter · Decision Letter 1]

31 Aug 2022

PONE-D-22-14548R1Relationship of spirituality, health engagement, health belief and attitudes toward acceptance and willingness to pay for a COVID-19 vaccinePLOS ONE

Dear Dr. Tsai,

Thank you for submitting your manuscript to PLOS ONE. After careful consideration, we feel that it has merit but does not fully meet PLOS ONE’s publication criteria as it currently stands. Therefore, we invite you to submit a revised version of the manuscript that addresses the points raised during the review process.

We look forward to receiving your revised manuscript.

Kind regards,

Harapan Harapan, MD, PhD

Academic Editor

PLOS ONE

Journal Requirements:

Reviewers' comments:

Reviewer's Responses to Questions

**Comments to the Author**

1. If the authors have adequately addressed your comments raised in a previous round of review and you feel that this manuscript is now acceptable for publication, you may indicate that here to bypass the “Comments to the Author” section, enter your conflict of interest statement in the “Confidential to Editor” section, and submit your "Accept" recommendation.

Reviewer #2: All comments have been addressed

2. Is the manuscript technically sound, and do the data support the conclusions?

Reviewer #2: Partly

3. Has the statistical analysis been performed appropriately and rigorously? 

Reviewer #2: Yes

4. Have the authors made all data underlying the findings in their manuscript fully available?

Reviewer #2: Yes

5. Is the manuscript presented in an intelligible fashion and written in standard English?

Reviewer #2: No

6. Review Comments to the Author

Reviewer #2: The writing has been improved. Authors answered all my concerns. Additional statistical assessments have improved the value of the manuscript. But, several minor concerns are left before the paper can be published.

1. WHO, COVID-19 and SARS-CoV-2 should be defined at their first appearance (especially in Introduction).

2. Also, regarding the abbreviations, author should keep this journal’s policy in mind: “Do not use non-standard abbreviations unless they appear at least three times in the text.” (https://journals.plos.org/plosone/s/submission-guidelines)

3. Line 61—63. “..1,816,041 people are reported to be infected, with 63 50,404 deaths [3].” As of when?

4. Line 63. “After scientists discovered this new SARS-CoV-2 strain…” I don’t think this is correct. Try this sentence, “After the viral sequence is published…”

5. Line 67—69 needs rephrasing with proper grammar.

6. Line 171—172. “the Cronbach's alpha value for spirituality was 0.70 that indicating acceptable reliability.” requires citation.

7. Phrases like “In the current study,” in Line 351 is not formal. We usually say, “In this present study,” or “Herein,”.

7. PLOS authors have the option to publish the peer review history of their article (what does this mean?). If published, this will include your full peer review and any attached files.

Reviewer #2: **Yes: **Muhammad Iqhrammullah

---

## [Author Response · Author response to Decision Letter 1]

6 Sep 2022

PONE-D-22-14548R2-Revision version 2

Response to Reviewer 2 Comments

Dear Reviewer #2,

Thank you for considering our manuscript and for the valuable suggestions, also the opportunity to resubmit a revised manuscript, which helps us to improve the article. We carefully revised the manuscript in accordance with your comments. The revised sections of the manuscript are marked with red color. Our point-by-point responses to the comments are as follows. We very much hope the revised manuscript is accepted for publication in PLOS ONE. Thank you very much for your consideration.

Point 1. WHO, COVID-19 and SARS-CoV-2 should be defined at their first appearance (especially in Introduction).

Response 1: Thank you for your valuable comment and suggestion. We defined COVID-19 at its first appearance. However, the WHO and SARS-CoV-2 do not use abbreviations because they appear only once in the text (please see line 59-64 on page 3).

Point 2. Also, regarding the abbreviations, author should keep this journal’s policy in mind: “Do not use non-standard abbreviations unless they appear at least three times in the text.” (https://journals.plos.org/plosone/s/submission-guidelines)

Response 2: Thank you for your valuable comment and suggestion. We revised the abbreviations to be explained at the beginning of the previous sentences follow the submission guidelines as follows: 

Abbreviations: 

Adjusted beta coefficients (�); Content validity index (CVI); Confidence intervals (CIs); Coronavirus disease 2019 (COVID-19); Health belief model (HBM); Health engagement (HE); Kaiser-Meyer-Olkin (KMO); Perceived barriers (PBA), Perceived benefits (PBE); Perceived severity (PSE); Perceived susceptibility (PSU), Standard deviation (SD), Vaccine attitudes (AVs).

Point 3. Line 61—63. “..1,816,041 people are reported to be infected, with 63 50,404 deaths [3].” As of when?

Response 3: Thank you for your insightful feedback. Based on the reviewer's advice, we have added the as of when to improve the presentation of the COVID-19 data as follows; (please see line 62—63 on page 3)

“From 3 January 2020 to 13 June 2021, this disease has spread to Indonesia, where approximately 1,816,041 people are reported to be infected, with 50,404 deaths”

Point 4. Line 63. “After scientists discovered this new SARS-CoV-2 strain…” I don’t think this is correct. Try this sentence, “After the viral sequence is published…”

Response 4: Thank you for your valuable suggestion. In order to make manuscripts to be better presented with precise, we re-organize the sentences based on the reviewer’s suggestion as follows; (please see line 64—65 on page 3)

“After the viral sequence (severe acute respiratory syndrome coronavirus 2) was published, vaccines for COVID-19 were rapidly developed to be distributed globally”

Point 5. Line 67—69 needs rephrasing with proper grammar.

Response 5: Thank you for your valuable suggestion. In order to make manuscripts to be proper grammar, we re-organize the sentences based on the reviewer’s suggestion as follows; (please see line 68—70 on page 3)

“Indonesia is unique because its citizens have higher positive spiritual beliefs related to health attitudes (Rias et al. 2020) and differences in health perspectives (Butandy et al. 2020), which may influence acceptance and willingness to pay for the COVID-19 vaccine.”

Point 6. Line 171—172. “the Cronbach's alpha value for spirituality was 0.70 that indicating acceptable reliability.” requires citation.

Response 6: Thank you for your insightful feedback. In this revised manuscript, we add the citation based on the literature presented that 0.70 indicates acceptable reliability as follows; (please see line 173—174 on page 7)

“The internal consistency reliability was calculated by Cronbach’s alpha test; a value ≥0.70 indicates acceptable reliability (De Vet et al. 2011)”

Reference: De Vet, H. C., Terwee, C. B., Mokkink, L. B., & Knol, D. L. (2011). Measurement in medicine: a practical guide. Cambridge University Press: Cambridge, UK.

Point 7. Phrases like “In the current study,” in Line 351 is not formal. We usually say, “In this present study,” or “Herein,”.

Response 7: Thank you for your valuable suggestion. We phrase “In the current study” to be “Herein” based on the reviewer’s suggestion (please see line 355, page 16).

thank you--------------------------------------------------

---

## [Editor Report · Decision Letter 2]

8 Sep 2022

Relationship of spirituality, health engagement, health belief and attitudes toward acceptance and willingness to pay for a COVID-19 vaccine

PONE-D-22-14548R2

Dear Dr. Tsai,

We’re pleased to inform you that your manuscript has been judged scientifically suitable for publication and will be formally accepted for publication once it meets all outstanding technical requirements.

Kind regards,

Harapan Harapan, MD, PhD

Academic Editor

PLOS ONE
---

## [Editor Report · Acceptance letter]

23 Sep 2022

PONE-D-22-14548R2 

Relationship of spirituality, health engagement, health belief and attitudes toward acceptance and willingness to pay for a COVID-19 vaccine 

Dear Dr. Tsai:

I'm pleased to inform you that your manuscript has been deemed suitable for publication in PLOS ONE. Congratulations! Your manuscript is now with our production department. 

Kind regards, 

on behalf of

Dr. Harapan Harapan 

Academic Editor

PLOS ONE